# Reduction of Cardiovascular Events and Related Healthcare Expenditures through Achieving Population-Level Targets of Dietary Salt Intake in Japan: A Simulation Model Based on the National Health and Nutrition Survey

**DOI:** 10.3390/nu14173606

**Published:** 2022-08-31

**Authors:** Nayu Ikeda, Hitomi Yamashita, Jun Hattori, Hiroki Kato, Katsushi Yoshita, Nobuo Nishi

**Affiliations:** 1International Center for Nutrition and Information, National Institute of Health and Nutrition, National Institutes of Biomedical Innovation, Health and Nutrition, Tokyo 162-8636, Japan; 2Department of Healthcare Information Management, The University of Tokyo Hospital, Tokyo 113-8655, Japan; 3Graduate School of Human Life and Ecology, Osaka Metropolitan University, Osaka 558-8585, Japan

**Keywords:** salt intake, blood pressure, cardiovascular disease, ischemic heart disease, stroke, simulation, Markov model, healthcare costs, incidence, mortality

## Abstract

Reducing population dietary salt intake is expected to help prevent cardiovascular disease and thus constrain increasing national healthcare expenditures in Japan’s super-aged society. We aimed to estimate the impact of achieving global and national salt-reduction targets (8, <6, and <5 grams/day) on cardiovascular events and national healthcare spending in Japan. Using published data including mean salt intake and systolic blood pressure from the 2019 National Health and Nutrition Survey, we developed a Markov model of a closed cohort of adults aged 40–79 years in 2019 (*n* = 66,955,000) transitioning among six health states based on the disease course of ischemic heart disease (IHD) and stroke. If mean salt intake were to remain at 2019 levels over 10 years, cumulative incident cases in the cohort would be approximately 2.0 million for IHD and 2.6 million for stroke, costing USD 61.6 billion for IHD and USD 104.6 billion for stroke. Compared with the status quo, reducing mean salt intake towards the targets over 10 years would avert 1–3% of IHD and stroke events and save up to 2% of related national healthcare costs. Attaining dietary salt-reduction goals among adults would yield moderate health economic benefits in Japan.

## 1. Introduction

Chronic exposure to a diet high in salt increases blood pressure and the risk for subsequent cardiovascular disease, chiefly ischemic heart disease (IHD) and stroke, which are the leading causes of death and disability worldwide [1]. The average dietary salt consumption of the global adult population in 2010 was estimated to be 10 grams per day [2], twice as high as the value recommended by the World Health Organization (WHO) of less than 5 grams per day [3]. A high salt intake ranked as a leading dietary risk factor for the global burden of noncommunicable diseases in 2017 [4].

Reduction of population-level dietary salt consumption is a priority intervention for the prevention and control of noncommunicable diseases through lowering blood pressure [5,6]. Salt reduction is also expected to ultimately save national healthcare expenditures by decreasing the need for the treatment of cardiovascular events. To project future health and economic impacts of interventions for dietary salt reduction, a number of modeling studies have been conducted in global and local settings [7,8,9,10,11,12,13,14,15,16]. Such information is increasingly important for the planning and execution of nutrition policies to improve public health.

In Japan, mean dietary salt intake decreased between the late twentieth and the early twenty-first centuries [17,18], but it remained at 9.7 grams per day among adults as of 2019 [19]. The incidence of stroke in Japan is the third highest among Organisation for Economic Co-operation and Development (OECD) member countries (after Latvia and Lithuania), and the incidence of IHD is approximately the OECD average [20]. The economic burden of cardiovascular disease is substantial, accounting for one-fifth of national healthcare expenditures [21]. To prevent cardiovascular disease and restrain future increases in national healthcare expenditures in Japan’s super-aged society, in which approximately 29% of the population is aged 65 years and older [22], a national health-promotion campaign in 2012 set a dietary salt-reduction goal for the Japanese population at 8 grams per day [23]. Dietary salt-reduction targets have also been recommended at <6 grams per day by the Japanese Society of Hypertension’s guidelines in 2019 [24] and <7.5 grams per day for men and <6.5 grams per day for women by the Dietary Reference Intakes for Japanese in 2020 [25]. However, the health economic consequences of nutrition policies, including such salt-reduction programs, have rarely been evaluated in Japan.

The objective of this study was to develop a simulation model for projecting the health and economic benefits of dietary salt reduction policies in Japan. Using a simulation model, we estimated the cardiovascular morbidity and mortality that would be prevented and the national healthcare expenditures that would be saved by achieving the global and national targets for dietary salt reduction in the adult population of Japan.

## 2. Materials and Methods

### 2.1. Modeling Framework and Study Population

We projected the effects of reducing population-level dietary salt intake to recommended targets on the incidence and mortality of IHD and stroke and national healthcare expenditures through lowering blood pressure. We defined IHD (I20–I25) and stroke (I60–I69) by the International Classification of Diseases, 10th Revision codes. We developed a discrete-time Markov cohort macro-simulation model by using TreeAge Pro Healthcare 2021 (TreeAge Software, Williamstown, MA, USA) [26]. We simulated a closed cohort of total population aged 40 to 79 years in Japan in 2019 (*n* = 66,955,000) [27] over 10 years. We excluded adults aged under 40 years due to low cardiovascular mortality in this age range [28] and those aged 80 years and older to avoid unstable estimates resulting from relatively small survey samples for dietary salt intake and systolic blood pressure (SBP) [19]. We selected 2019 as the baseline year of simulation because it was the most recent year having all the data available for the input parameters described below. We restricted the time horizon to 10 years to ensure the robustness of the study results without being influenced by potential long-term social changes that might occur. We divided the simulation period into annual cycles and conducted the simulation from the perspective of the healthcare sector. We stratified the Markov model by sex and 10-year age group.

The model consisted of six mutually exclusive health states: being healthy, chronic IHD, chronic stroke, death from IHD, death from stroke, and death from other causes (Figure 1). The healthy state represented people who had no history of IHD or stroke. The states of chronic IHD and chronic stroke had survivors of each disease who continued a treatment on an outpatient basis. The three death states were absorbing terminal states.

At the beginning of the Markov process, the cohort was split among the three health states of being healthy, chronic IHD, and chronic stroke on the basis of the prevalence of IHD and stroke. In each successive year, people in the cohort moved among the six health states according to annual transition probabilities determined by incidence, acute case fatality, and mortality. People in a healthy state either maintained good health, developed a first-ever IHD or stroke event, or died from other causes; they could not return to the healthy state after leaving with an occurrence of IHD or stroke. Outpatients in the chronic states either stayed in the same condition, experienced a recurrent event, or died from other causes. Following the onset of a first-ever or recurrent event, patients exited the healthy or chronic states upon receiving inpatient care in the acute phase. Acute patients entered the states of IHD and stroke death if they died from the disease by the end of the current cycle, or they moved to the chronic states if they survived to discharge for continuing outpatient care in the next annual cycle. To simplify the model, we made two assumptions according to previous studies: (1) mortality rates from other causes were equal among the healthy and chronic states [12]; and (2) the full effect of dietary salt reduction on IHD and stroke was achieved immediately without any phase-in period [10,11].

### 2.2. Scenarios

We considered three scenarios of reducing mean dietary salt intake towards global and national targets over 10 years: 8 grams per day as set by the national health promotion campaign [23]; <6 grams per day as recommended by the Japanese Society of Hypertension’s guidelines [24]; and <5 grams per day as recommended by the WHO guidelines [3]. For assessing health gains and savings of national healthcare expenditures under these scenarios, we set the base-case scenario of mean dietary salt consumption staying at the levels of 2019 as a comparator. We adopted this reference scenario of the status quo, considering that mean dietary salt intake had remained virtually constant for the previous five years in the Japan National Health and Nutrition Surveys [19]. We assessed the health and economic impacts of reducing dietary salt intake as differences in projected values of the incidence, mortality, and national health expenditures between the base-case scenario and each scenario of dietary salt reduction.

We supposed that mean dietary salt intake decreased annually at a constant rate under each scenario so that age-adjusted mean dietary salt intake would reach the targets in 2029. We used the standard population of Japan in 2010 [29] for age adjustment. We calculated annual reduction rates in mean dietary salt intake at 2.1% for 8 grams per day, 4.9% for <6 grams per day, and 6.8% for <5 grams per day across all sex and age groups. Mean dietary salt intake to be attained over time under each scenario is illustrated in Appendix A.

### 2.3. Input Parameters

We obtained data on input parameters from existing databases, survey reports, and previous studies (Table 1).

We obtained data on mean dietary salt intake and mean SBP from the Japan National Health and Nutrition Survey in 2019 [19] (Appendix A). This survey used a stratified two-stage cluster sample design to obtain a nationally representative sample of the non-institutionalized Japanese population [36]. Household representatives reported food intake of household members with a 1-day semi-weighted household dietary record. The Standard Tables of Food Composition in Japan 2015 [37] was applied to quantify salt intake of participants from their food intake records. At a physical examination site, health professionals used a Riva-Rocci mercury manometer to measure blood pressure twice in the right upper arm of seated participants after 5 min of rest. Population-level mean SBP was estimated from averages of two readings or single readings if measured only once on individual participants.

We multiplied proportions of first-ever events (Appendix A) with the incidence to divide incident cases in the cohort into first-ever events in the healthy state and recurrent ones in the chronic states. We used 28-day case fatality rates (Appendix A) as transition hazards from an incident IHD and stroke to death.

We assumed that the cardiovascular health benefits of salt reduction were mediated through blood pressure reductions. To calculate changes in mean SBPs associated with reductions in salt intake, we applied epidemiological associations estimated from previous regression analyses using the same 10-year age groups with those in the present study [33] (Appendix A). Mean SBP to be expected over time under each scenario is demonstrated in Appendix A. We used relative risks for IHD and stroke associated with raised SBP (Appendix A) to quantify changes in the incidence of IHD and stroke through decreased mean SBP. We supposed that lowered SBP through dietary salt reduction would affect incidence but would not impact survival after the onset.

Regarding national healthcare expenditures (Appendix A), we assigned the sum of outpatient care costs and drug prescription costs to the transitions of staying in the chronic IHD and stroke and inpatient care costs to the transitions of developing first-ever and recurrent events. We converted national healthcare expenditures from Japanese yen to U.S. dollars (USD) according to the annual average exchange rate in 2019 published by the International Monetary Fund (109.01 yen per U.S. dollar) [38]. We discounted national healthcare expenditures at 2% annually, according to the guidelines for economic evaluation of healthcare technologies in Japan [39].

### 2.4. Model Validation and Sensitivity Analyses

We validated the model by checking if it reproduced national healthcare expenditures for IHD and stroke among the population aged 40 to 79 years in 2019. We compared actual data to projected national healthcare expenditures for outpatient and inpatient care in the first annual cycle of the base-case scenario.

We conducted multiple deterministic one-way sensitivity analyses to assess the impact of uncertainty surrounding model input parameters on expected savings of national healthcare expenditures. Parameters examined in the sensitivity analyses were the discount rate (0–4%), associations between salt intake and SBP, and the incidence, prevalence, and relative risks of IHD and stroke (lower and upper bounds of 95% confidence intervals in Appendix A).

## 3. Results

### 3.1. Projected Incidence, Mortality, and National Healthcare Expenditures on the Base Case

In the model validation, projected national healthcare expenditures on outpatient and inpatient care for IHD and stroke in the first annual cycle under the base-case scenario equaled actual values in 2019 across all age groups and both sexes. We therefore considered our simulation model to be sufficiently accurate to reproduce the baseline national healthcare expenditures.

Table 2 shows the projected cumulative total of incidence, mortality, and national healthcare expenditures for IHD and stroke at the end of the simulation period under the base-case scenario. If mean dietary salt intake was maintained at the baseline levels of 2019 over a decade, the number of first-ever and recurrent incident cases in the cohort would total 2,000,792 (3.0%) for IHD and 2,595,308 (3.9%) for stroke. The number of accumulated deaths would reach 747,314 (1.1%) for IHD and 398,255 (0.6%) for stroke. The cumulative total of discounted national healthcare expenditures would amount to USD 61.6 billion for IHD (USD 36.2 billion for outpatient care and USD 25.4 billion for inpatient care) and USD 104.6 billion for stroke (USD 43.1 billion for outpatient care and USD 61.5 billion for inpatient care).

### 3.2. Health Gains by Achieving Dietary Salt Reduction Targets

Figure 2 and Figure 3 demonstrate cumulative totals of incidence and mortality prevented through achieving population-level dietary salt-reduction targets over 10 years compared with the base-case scenario. If mean dietary salt intake declined to 8, <6, and <5 grams per day over 10 years, the accumulated total number of averted incident cases in the whole cohort would be 51,201 (1.1%; 21,594 for IHD and 29,607 for stroke), 106,055 (2.3%; 44,852 for IHD and 61,203 for stroke), and 134,633 (2.9%; 50,734 for IHD and 77,599 for stroke), respectively. Relative reductions of incident cases of both IHD and stroke under the salt targets of 8, <6, and <5 grams per day were larger in men at 1.2%, 2.5%, and 3.1%, respectively, than in women at 1.0%, 2.1%, and 2.7%, respectively.

The cumulative total number of averted deaths in the whole cohort was projected to be 12,524 (1.1%; 7990 for IHD and 4534 for stroke) for 8 grams per day, 25,964 (2.3%; 16,593 for IHD and 9371 for stroke) for <6 grams per day, and 32,979 (2.9%; 21,097 for IHD and 11,882 for stroke) for <5 grams per day. Relative reductions of deaths from both IHD and stroke under the salt targets of 8, <6, and <5 grams per day were larger in men at 1.2%, 2.4%, and 3.1%, respectively, than in women at 1.0%, 2.1%, and 2.6%, respectively.

### 3.3. National Healthcare Expenditures Saved by Achieving Dietary Salt Reduction Targets

Figure 4 presents the projected cumulative total savings of national healthcare expenditures through the prevention of IHD and stroke over 10 years under the scenarios of achieving dietary salt-reduction targets. The accumulated total savings in the whole cohort would amount to USD 1.2 billion (0.7%; USD 363 million for IHD and USD 789 million for stroke) for 8 grams per day, USD 2.4 billion (1.4%; USD 758 million for IHD and USD 1.6 billion for stroke) for <6 grams per day, and USD 3.1 billion (1.8%; USD 967 million for IHD and USD 2.1 billion for stroke) for <5 grams per day. Relative reductions of national healthcare expenditures under the salt targets of 8, <6, and <5 grams per day were larger in men at 0.8%, 1.6%, and 2.0%, respectively, than in women at 0.6%, 1.2%, and 1.6%, respectively.

### 3.4. Sensitivity Analyses

Figure 5 illustrates the results of the one-way sensitivity analyses in the whole cohort. The modeled uncertainty in cumulative savings of national healthcare expenditures over 10 years was largest for the changes in SBP associated with changes in salt intake and the relative risks for stroke associated with SBP, followed by relative risks for IHD or the discount rate.

## 4. Discussion

To our knowledge, this is the first simulation study on the health economic effects of nutrition policies including salt intake-reduction policies in Japan. The global burden of disease attributable to dietary risks is substantial [40], and previous studies have projected future reductions in disability-adjusted life years through increased fruit and vegetable intake [41,42,43] and decreased salt intake [44] in Japan. However, the effects of healthy diet on the control of national healthcare expenditures were not well-understood. Our simulation of medical cost savings through population dietary salt reduction provides useful information for strengthening nutrition policies to suppress the trends toward increased national healthcare expenditures in Japan’s super-aged society.

Our model integrated the latest information on key input parameters available from official statistics and the published literature, especially mean dietary salt intake and SBP from the Japan National Health and Nutrition Survey. Our results suggest that achieving the global and national dietary salt-reduction targets among adults in Japan would have moderate effects on the control of cardiovascular disease and related national healthcare expenditures through lowering population-level blood pressure. In the cohort aged 40 to 79 years in 2019, compared with the status quo, a reduction of mean dietary salt intake to 8, <6, and <5 grams per day over 10 years would avert approximately 1 to 3 percent of IHD and stroke events and save up to 2 percent of national healthcare expenditures for cardiovascular disease.

Largely because of the absence of a national database on unit costs of nutrition policies, our preliminary study focused on estimating the health economic benefits of achieving dietary salt-reduction targets. We explored relatively simple scenarios of reducing population dietary salt intake towards targets without specifying any actual intervention program. Similar investigations have been conducted on reductions of mean dietary salt intake by 1 to 3 grams per day in Finland [11] and the United States [9] as well as reductions to national goals of 6 and 9 grams per day in China [13] and 7 and 8 grams per day in Vietnam [15]. These previous studies applied a Markov cohort model, notably the Coronary Heart Disease Policy model [9,13], and used summary statistics on dietary salt intake and blood pressure obtained from national health and nutrition surveys. Their results agreed with ours in Japan in that the health economic effects of reducing population dietary salt intake were moderate.

Projected health gains and cost savings through dietary salt reduction were slightly greater in men than in women. This result was consistent with a previous finding in Vietnam [15]. The benefits of population salt reduction were greater for men than women in Japan, partly because annual decreases in mean dietary salt intake were set in the simulation model to be proportional to the baseline levels. A larger decrease in mean dietary salt intake was required to meet the national targets in men than in women.

The strengths of our study should be addressed. First, we used recent summary statistics on salt intake and SBP at the population level from the National Health and Nutrition Survey. Second, we validated the model, including its assumptions by comparison of baseline results to actual healthcare expenditures. Third, we conducted one-to-one sensitivity analyses to explore the uncertainty of results.

Our study also has several limitations to be noted. First, we did not examine the cost-effectiveness of specific intervention programs for population dietary salt reduction, such as mass media campaigns, front-of-package nutrition labeling, and reformulation of processed foods. To facilitate implementation of such cost-effectiveness evaluations, building a national network database of unit costs and use of nutritional interventions is an urgent need. Second, while focusing on medical expenditures on inpatient and outpatient treatment, our model did not incorporate disability requiring long-term care in the community or clinical settings. Preventing long-term care needs, particularly among stroke survivors in the older population, would have considerable economic implications in Japan. Third, our model on the closed cohort aged 40 to 79 years in 2019 ignored the entry of people reaching 40 years of age and the exit of those surviving to 80 years of age during the simulation period. Our results should be interpreted as hypothetical projections of that specific cohort over a decade. Fourth, our model on IHD and stroke did not cover other diseases related to excessive salt intake, such as hypertensive heart disease, chronic kidney disease, and gastric cancer. Different model structures may be needed to adjust for disease processes that are different from those of IHD and stroke adopted in our model involving acute and chronic phases for the first and recurrent incidents. In addition, although our model included deaths from other causes, it did not explicitly account for competing risks of death, relying on the prevalence, incidence, mortality, and relative risks of IHD and stroke estimated by the Global Burden of Disease study. Fifth, salt intake in 2019 was calculated using the Standard Table of Food Composition in 2015. Food reformulation during the four intervening years might result in overestimation of salt intake although there is insufficient information available to assess the impact.

Despite the decline in dietary salt intake in the Japanese population from the second half of the twentieth century, progress has stalled, and national healthcare spending on cardiovascular disease has been on the rise in this super-aged society. Our findings support calls for reinforcing population-wide interventions to leverage dietary salt intake reductions from an economic standpoint. Both supply- and demand-side approaches would be necessary for addressing the primary source of dietary salt in Japanese adults, i.e., seasonings such as salt used in cooking and at the table, soy sauce, miso (soybean paste), and soup stock [45].

In future research, the simulation model should be extended to cover the health economic effects of population dietary salt intake reductions on disability, quality of life, and use of long-term care services over a lifetime. Prevention of long-term care needs among dependent people would be crucial for ensuring economic benefits from additional life years gained through reduced cardiovascular mortality [46]. In addition, the impacts on overall healthcare expenditures should be explored, considering morbidities that are likely to occur instead of IHD and stroke in prolonged lifespan. Moreover, the projected health economic benefits of dietary salt reduction should be compared with those of other strategies for lowering blood pressure under a common framework. A previous regression analysis of the Japan National Nutrition Surveys between 1986 and 2002 suggested that contributions to the decline of mean SBP were limited for small reductions of dietary salt intake compared with substantially increased use of antihypertensive medications in older adults and lowered body weight in young women [47]. Such findings support the need for comparative simulation studies that would enhance public understanding of the relative importance of the population’s salt reduction in the control of future national healthcare expenditures and social security costs.

To conclude, this first simulation model of future health economic consequences of dietary salt-reduction policies in Japan suggests that accomplishing global and national dietary salt-reduction targets would save up to 2% of national healthcare expenditures for cardiovascular disease among middle-aged and older adults over a decade. Health economic assessments of nutrition policies via simulation models should be further encouraged under the principles of sustainable healthy diets [48] and the planetary health diet [49] to realize a truly healthy and long-lived society.

## Figures and Tables

**Figure 1 nutrients-14-03606-f001:**
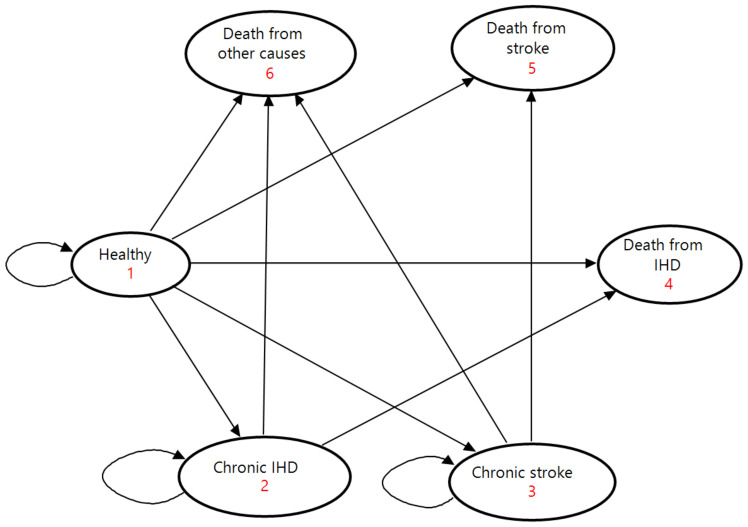
State-transition diagram of the Markov model. Ovals represent the six health states. Direct arrows represent directions of transitions of the cohort between health states. Circular arrows indicate the cohort remaining in each health state. Each arrow has a transition probability. IHD, ischemic heart disease.

**Figure 2 nutrients-14-03606-f002:**
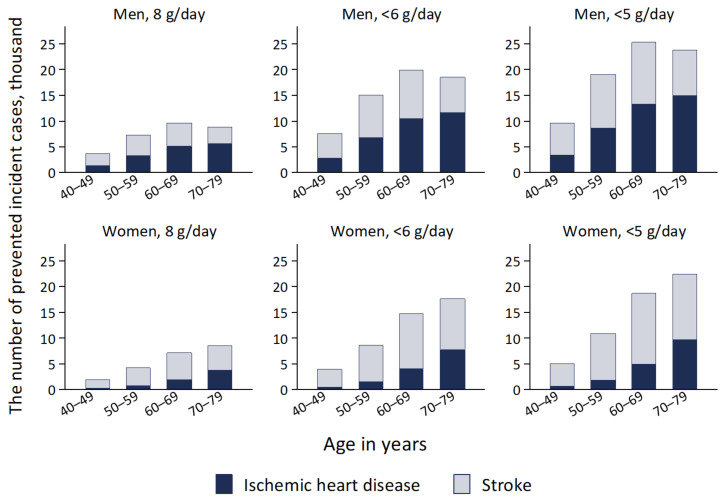
Projected cumulative totals of incident cases from ischemic heart disease and stroke prevented by reducing mean dietary salt intake to 8 g/day, <6 g/day, and <5 g/day between the years 2019 and 2029 in a closed cohort of population aged 40–79 years in 2019.

**Figure 3 nutrients-14-03606-f003:**
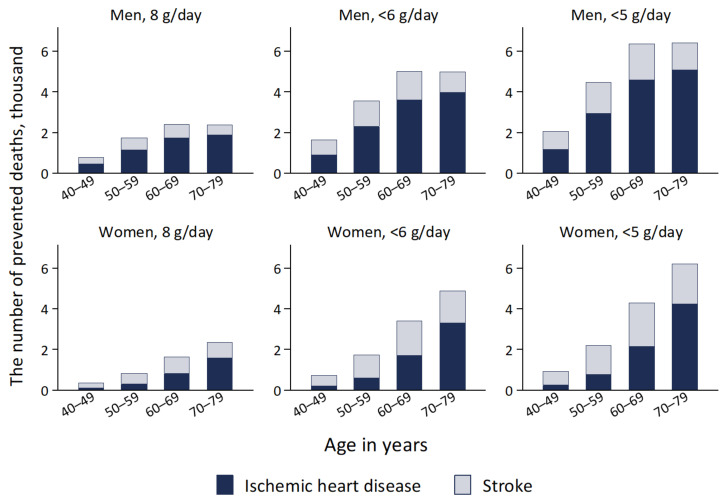
Projected cumulative prevented deaths from ischemic heart disease and stroke by reducing mean dietary salt intake to 8 g/day, <6 g/day, and <5 g/day between the years 2019 and 2029 in a closed cohort of population aged 40–79 years in 2019.

**Figure 4 nutrients-14-03606-f004:**
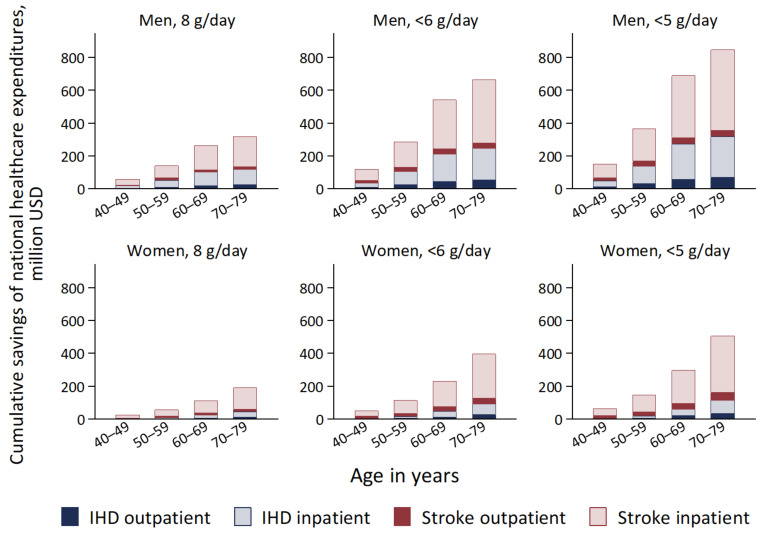
Projected cumulative total savings of national healthcare expenditures for IHD and stroke by reducing mean dietary salt intake to 8 g/day, <6 g/day, and <5 g/day between the years 2019 and 2029 compared with the base-case scenario in a closed cohort of population aged 40–79 years in 2019. IHD, ischemic heart disease.

**Figure 5 nutrients-14-03606-f005:**
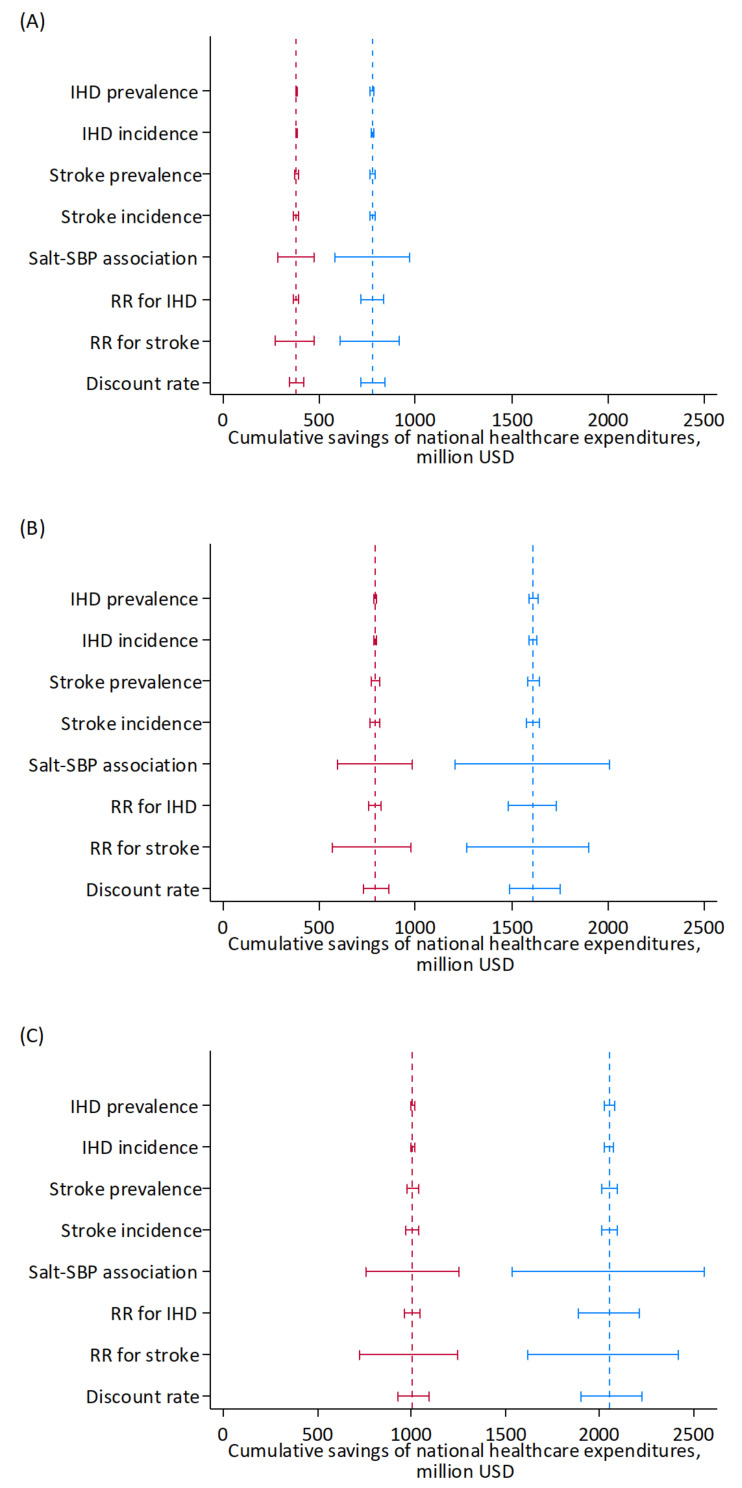
Results of the one-way sensitivity analyses of key input parameters on projected cumulative savings of national healthcare expenditures for IHD and stroke by reducing mean dietary salt intake to 8 g/day (**A**), <6 g/day (**B**), and <5 g/day (**C**) between the years 2019 and 2029 in a closed cohort of population aged 40–79 years in 2019. Blue, men; red, women. Range plots with spikes indicate uncertainty ranges. Dashed lines indicate expected values. IHD, ischemic heart disease; RR, relative risk, SBP: systolic blood pressure.

**Table 1 nutrients-14-03606-t001:** Input parameters and data sources used in the Markov model.

Input Parameters	Data Sources	Values
Total population	Population Estimates [27]	Appendix A
Mean dietary salt intake	National Health and Nutrition Survey in Japan, 2019 [19]	Appendix A
Mean SBP	National Health and Nutrition Survey in Japan, 2019 [19]	Appendix A
Prevalence rates of IHD and stroke	Global Burden of Disease Study 2019 [20]	Appendix A
Incidence rates of IHD and stroke	Global Burden of Disease Study 2019 [20]	Appendix A
Mortality rates of IHD, stroke, and all causes	Global Burden of Disease Study 2019 [20]	Appendix A
Proportion of first-ever events in incident cases		
IHD	Japan Thrombosis Registry for Atrial Fibrillation, Coronary, or Cerebrovascular Events (J-TRACE) [30]	Appendix A
Stroke	Shiga Stroke Registry [31]	Appendix A
28-day case fatality rates of IHD and stroke	Takashima Cardiovascular Disease Registration System [32]	Appendix A
Changes in SBP associated with changes in salt intake	Analysis of observational studies in 24 communities [33]	Appendix A
Relative risks for IHD and stroke associated with SBP	Global Burden of Disease Study 2019 [20]	Appendix A
National healthcare expenditures	Survey on Medical Care Benefit, 2019 [34], Survey on Prescription Drug Expenditure, 2019 [35]	Appendix A

IHD, ischemic heart disease; SBP, systolic blood pressure.

**Table 2 nutrients-14-03606-t002:** Projected cumulative incidence, mortality, and national healthcare expenditures for ischemic heart disease (IHD) and stroke over 10 years from 2019 under the base-case scenario of mean dietary salt intake remaining at the levels of 2019.

Sex, Age (Years)	Population, 2019	Incidence				Deaths				National Healthcare Expenditures, Million USD			
		IHD		Stroke		IHD		Stroke		Outpatient		Inpatient	
	No.	No.	(%)	No.	(%)	No.	(%)	No.	(%)	IHD	Stroke	IHD	Stroke
Men													
40–79	32,795,000	1,322,550	(4.0)	1,151,035	(3.5)	453,635	(1.4)	171,504	(0.5)	24,671	21,160	20,001	36,052
40–49	9,373,000	125,027	(1.3)	201,224	(2.1)	42,884	(0.5)	29,982	(0.3)	1504	2060	1314	2798
50–59	8,160,000	263,831	(3.2)	292,985	(3.6)	90,494	(1.1)	43,655	(0.5)	3560	3814	3227	5418
60–69	7,930,000	403,995	(5.1)	340,586	(4.3)	138,570	(1.7)	50,747	(0.6)	7642	6298	6416	10,520
70–79	7,332,000	529,697	(7.2)	316,240	(4.3)	181,686	(2.5)	47,120	(0.6)	11,965	8989	9043	17,317
Women													
40–79	34,160,000	678,243	(2.0)	1,444,273	(4.2)	293,679	(0.9)	226,751	(0.7)	11,514	21,908	5448	25,479
40–49	9,147,000	25,083	(0.3)	170,857	(1.9)	10,861	(0.1)	26,825	(0.3)	603	1855	160	1842
50–59	8,118,000	64,238	(0.8)	291,997	(3.6)	27,815	(0.3)	45,844	(0.6)	1174	3608	439	3267
60–69	8,302,000	171,860	(2.1)	443,201	(5.3)	74,415	(0.9)	69,582	(0.8)	2953	6421	1410	6252
70–79	8,593,000	417,062	(4.9)	538,218	(6.3)	180,588	(2.1)	84,500	(1.0)	6783	10,024	3439	14,118

## Data Availability

All data supporting this research are provided in this manuscript.

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
