# Peer review of "Reduction of Cardiovascular Events and Related Healthcare Expenditures through Achieving Population-Level Targets of Dietary Salt Intake in Japan: A Simulation Model Based on the National Health and Nutrition Survey"

_nutrients, 2022, doi:10.3390/nu14173606_

Round 1

Reviewer 1 Report

The authors Ikeda et al. present their finding from a simulation study evaluating the health and healthcare expenditure impacts of dietary salt reduction towards national and WHO target levels in Japan. The paper is well written and mostly easy to read. The report modest impacts on CVD IHD and stroke events and deaths as well as health expenditure savings. I have some comments for the authors to consider.

1. It appears the model is mainly a deterministic model? Results are presented without uncertainty around the point estimates. It might be good to see probabilistic sensitivity analysis to see the combined impact of uncertain model input parameters on the outcomes.

2. It was good to see the one-way sensitivity analysis. However, another important determinant in the model, the salt/sodium-SBP relationship was not assessed. It would be good to see the impact of the uncertainty in this relationship on the results. Furthermore, to quantify this salt-SBP relationship, authors seem to rely on regression estimates that are somewhat dated. There is recent evidence from meta-analyses of long-term population trials that could be used. E.g. Huang et al. (https://www.bmj.com/content/368/bmj.m315).

3. Healthcare expenditure: The current presentation (in Table S5) could be improved if authors add provide cost per case of IHD and for stroke. Secondly, the focus appears to be mainly on CVD-related costs? Strategies that prolong life, are likely to lead to health expenses from other causes. This is of particular relevance to Japan, which the authors repeatedly highlight has an ageing population. With age comes co- and multi-morbidity, and hence healthcare costs. Did authors take into account the impact on overall health expenditure in their estimates? It would be good to see a discussion on this.

4. Other blood pressure-related CVDs e.g. hypertensive heart disease, CKD, etc. were not included, as well as the impact on stomach cancer. It might be good to discuss these, too.

5. Can author discuss how they handled competing risk of death?

Reviewer 2 Report

It is an interesting paper with a clear purpose. The paper is well-written. Model and assumptions are well described. Good to see model was validated by comparing baseline results to real expenditures in 2019.

Though, I still have a few minor comments.

page 2, line 77: authors mention those aged 80 years and older are excluded to ensure stability of estimates. This is not clear to me. Could they please elaborate a bit more on this?

Page 3, line 108 authors mention 2 assumptions made, but I don't see a reference or justification for these these assumption in the discussion.

Page 4, line 140: Intakes in 2019 were calculated using a food composition table from 2015. That means a 4-year gap and foods could have been reformulated already, which could imply an overestimation of the intake? This could be addressed in the discussion.

Discussion: The authors do describe the limitations of the study but could elaborate a bit more on the strengths of the study; model validated, sensitivity analyses performed to explore uncertainty. They could also come back on some assumptions made like mentioned earlier (maybe it is as simple as that the validation shows assumptions made are valid).

page 11: line 317: authors do refer to as study comparing analysis from 1886 and 2002, but not what the result was and what this would mean. Could they please elaborate a bit more?
